# Anchor-Final Self-Supervision Drives Hallucination-Aware Optimization in Large Vision-Language Models

**Jiaxi Liu**[1]  **Yifeng Yang**[1]  **Xinbing Wang**[1]  **Qinying Gu**[2]  **Nanyang Ye**[1 3 2]

## Abstract

Hallucinations in large vision-language models (LVLMs) remain a critical challenge, where models often generate tokens that fail to align with visual evidence. To address this issue, we propose AFS: Anchor-Final Self-Supervision, a novel framework for hallucination-aware optimization in LVLMs. By leveraging discrepancies between intermediate and final layer predictions, AFS selectively applies self-supervision to visually descriptive tokens, incorporates hallucination-aware token classification, and encourages consistency between intermediate and final layer distributions. Unlike traditional methods that rely on explicit supervision or post-hoc interventions, AFS optimizes the model via Group Relative Policy Optimization (GRPO), using token-specific rewards derived from internal model signals. Experiments demonstrate that AFS significantly reduces hallucinations without compromising recall in caption generation. Beyond captioning, AFS excels in discriminative tasks, improving the reliability of object existence predictions and multimodal reasoning. Furthermore, AFS demonstrates strong cross-dataset generalization, transferring effectively across diverse visual domains. Code is available at `https://github.com/guavayew/AFS`.

## 1. Introduction

Large Vision-Language Models (LVLMs) (Wang et al., 2024b; Lu et al., 2024; Liu et al., 2024a; Yao et al., 2024) have made significant progress across a wide range of multimodal tasks, including image captioning (Rohrbach et al.,

[1]Shanghai Jiao Tong University, Shanghai, China [2]Shanghai Artificial Intelligence Laboratory, Shanghai, China [3]Shanghai Innovation Institute, Shanghai, China. Correspondence to: Nanyang Ye <ynylincoln@sjtu.edu.cn>.

*Proceedings of the 43rd International Conference on Machine Learning*, Seoul, South Korea. PMLR 306, 2026. Copyright 2026 by the author(s).

2018), visual question answering (Li et al., 2023b), and open-ended visual reasoning (Fu et al., 2025), while also showing strong potential in real-world applications such as autonomous driving (Zhou et al., 2024a) and embodied AI (Ma et al., 2026). Despite their impressive generative capabilities, LVLMs are still plagued by hallucinations, where generated descriptions contain objects, attributes, or relations that are either unsupported or contradicted by the input image (Bai et al., 2024). Such hallucinations undermine the reliability of LVLMs, particularly in safety-critical applications, highlighting the importance of mitigating hallucinations in the pursuit of trustworthy multimodal artificial intelligence (Zhang et al., 2024c).

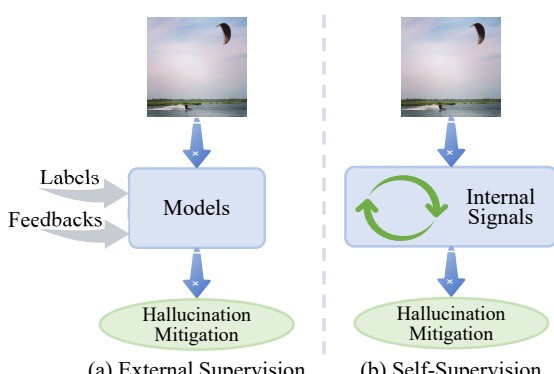

(a) External Supervision  (b) Self-Supervision

*Figure 1.* Comparison of training-time hallucination mitigation methods. **Left:** Traditional method with external supervision using labels, annotations and feedback. **Right:** Our method with self-supervision using internal model signals to reduce hallucinations.

Existing hallucination mitigation methods mainly fall into two categories. As shown in Figure 1, training-time approaches typically reduce hallucinations by strengthening supervision signals, such as constructing instruction-tuning datasets with carefully designed negative samples (Liu et al., 2023; Chen et al., 2025), collecting fine-grained hallucination annotations (Zhang et al., 2024a), or aligning models through contrastive learning (Sarkar et al., 2024; Li et al., 2025) and preference optimization (Sun et al., 2024; Zhou et al., 2024b). Although effective, these methods often rely on ground-truth captions, explicit labels, human feedback, or external evaluators, resulting in increased data costs and

limited scalability. In contrast, decoding-time approaches intervene during inference by adjusting token selection, attention patterns, or output distributions to suppress hallucinated predictions (Huo et al., 2024; Gong et al., 2024; Deng et al., 2024; Favero et al., 2024; Zhang et al., 2024b). Although attractive due to their training-free nature, these methods operate only post-hoc and are constrained by frozen model parameters, leaving the model's internal tendency to favor language priors over visual evidence unchanged, thus hindering their ability to generalize well across different prompts or domains.

Despite the progress made by both training-time and decoding-time approaches, a common limitation remains: hallucinations are mainly mitigated through external supervision or post-hoc corrections, rather than by directly engaging with the internal generation dynamics through which hallucinations arise. Recent analyses have highlighted that hallucinations in LVLMs arise not merely as output-level errors, but as a process-level issue stemming from the imbalance between visual evidence and dominant language priors during autoregressive generation (Liu et al., 2024c; Hu et al., 2025). In particular, visually unsupported tokens often receive minimal support in the intermediate layers but are amplified during later stages of decoding, where linguistic priors increasingly dominate the final predictions (Wang et al., 2024a; Jiang et al., 2025). This layer-wise discrepancy indicates that hallucinations are encoded in the model's internal probability dynamics, rather than arising solely from decoding choices. These observations motivate a different perspective on hallucination mitigation: can hallucinations be addressed during training by leveraging this layer-wise discrepancy as a self-supervised signal to optimize the internal generation dynamics?

To explore this idea, we propose AFS: Anchor-Final Self-Supervision, a novel training-time framework that operates on internal model signals. By treating hallucination mitigation as a token-aware optimization problem, our method enables more precise control over the generation process. Specifically, our method consists of three key components: (i) visual-selective reward gating, which restricts optimization to visually descriptive tokens while leaving non-visual generation unaffected; (ii) hallucination-aware token classification, which distinguishes hallucinated and visually grounded tokens based on anchor layer confidence and anchor-final probability gaps; and (iii) an anchor-final distributional consistency reward that penalizes late-stage amplification inconsistent with intermediate visual evidence. We apply Group Relative Policy Optimization (GRPO) to fine-tune the model during training, with these components collaborating to optimize the generation process. Through extensive experiments, we demonstrate that AFS effectively reduces hallucinations without sacrificing recall, enhances the model's ability to ground visual content across both

generative and discriminative tasks, and achieves strong generalization across diverse visual domains. Our main contributions are summarized as follows:

- We introduce a self-supervised framework for hallucination mitigation in LVLMs that operates primarily on internal model signals, without the need for ground-truth captions, preference labels, or external evaluators.

- We propose a token-aware optimization strategy that integrates visual-selective gating, hallucination-aware token classification and anchor-final distributional consistency reward to effectively mitigate hallucinations.

- Extensive experiments demonstrate that our method significantly reduces hallucinations in both generative and discriminative tasks, while also enhancing generalization across diverse datasets.

## 2. Preliminaries

**Group Relative Policy Optimization**    Group Relative Policy Optimization (GRPO) (Shao et al., 2024) is a policy optimization framework that eliminates the need for an explicit value function by estimating advantages through relative comparisons within a group of sampled outputs. For a given input query $q$, the behavior policy $\pi_{\theta_{\text{old}}}$ samples a group of responses $\{o_i\}_{i=1}^G \sim \pi_{\theta_{\text{old}}}(\cdot \mid q)$, each response $o_i$ is assigned a scalar reward $r_i = r(q, o_i)$. The group-relative advantage for the $i$-th response is computed by normalizing its reward within the group:

$$A_i = \frac{r_i - \text{mean}(\{r_j\}_{j=1}^G)}{\text{std}(\{r_j\}_{j=1}^G)}. \quad (1)$$

GRPO optimizes the following objective:

$$J_{\text{GRPO}}(\theta) = \mathbb{E}_{q,\{o_i\}_{i=1}^G} \left[ \frac{1}{G} \sum_{i=1}^G \left( \frac{1}{|o_i|} \sum_{t=1}^{|o_i|} \min \left( w_{i,t}(\theta) A_i, \right. \right. \right.$$

$$\left. \left. \left. \text{clip}(w_{i,t}(\theta), 1 - \epsilon, 1 + \epsilon) A_i \right) - \beta D_{\text{KL}}(\pi_\theta \| \pi_{\text{ref}}) \right) \right], \quad (2)$$

where the importance sampling ratio is defined as:

$$w_{i,t}(\theta) = \frac{\pi_\theta(o_{i,t} \mid q, o_{i,<t})}{\pi_{\theta_{\text{old}}}(o_{i,t} \mid q, o_{i,<t})}. \quad (3)$$

Here, $\epsilon$ and $\beta$ are hyperparameters controlling the clipping range and the strength of KL regularization, respectively.

**Why GRPO for AFS?**    We leverage GRPO to optimize hallucination mitigation because it enables adaptive optimization using token-aware rewards derived from internal model signals. In our self-supervised framework, hallucinations are mitigated by differentiating the quality of generated

tokens based on the anchor-final probability gap at the token level and anchor-final distributional consistency at the layer level. GRPO's reward mechanism is well-suited for this, as it enables the model to adjust its token generation process by evaluating these internal metrics and assigning different rewards, ensuring that predictions align with both intermediate and final layers. This makes GRPO an ideal choice for our method, as it facilitates nuanced optimization without requiring external supervision or ground-truth captions.

## 3. Method

### 3.1. Problem Formulation

Let $x$ denote an input image, $q$ denote the input instruction, and $y = (y_1, \ldots, y_T)$ be a response sequence generated autoregressively by a vision-language model $\pi_\theta$. At each generation step $t$, the model predicts a conditional distribution over the vocabulary $\mathcal{V}$ given the image, instruction, and previously generated tokens, from which the next token is sampled:

$$y_t \sim \pi_\theta(\cdot \mid x, q, y_{<t}), \qquad (4)$$

where $y_{<t} = (y_1, \ldots, y_{t-1})$.

Our objective is to assign a token-level reward $r_t$ to each generated token $y_t$ that (i) selectively targets visually descriptive tokens, (ii) differentiates hallucinated tokens from visually grounded ones using internal model signals, and (iii) enforces distributional consistency between intermediate and final layer predictions. To achieve this, we propose AFS, a self-supervised token-aware optimization framework integrating three tightly coupled components: visual-selective reward gating, hallucination-aware token classification based on anchor-final probability gaps, and an anchor-final distributional consistency reward that penalizes late-stage amplification inconsistent with intermediate visual evidence, as illustrated in Figure 2. We next describe each component in detail.

### 3.2. Visual-Selective Reward Gating

Hallucinations in LVLMs often arise from an imbalance between visual evidence and dominant language priors during autoregressive generation (Liu et al., 2024c; Hu et al., 2025). Under this imbalance, models may generate visually descriptive tokens with high confidence despite weak or absent visual support from the image, instead being driven by spurious correlations in the linguistic context (Han et al., 2022; Wu et al., 2022; Yan et al., 2023; Zhibo et al., 2023). Prior training-time approaches typically employ sequence-level or token-agnostic supervision signals. However, such objectives fail to decouple visual grounding from language modeling, since modifying functional or syntactic tokens does little to correct hallucinated visual predictions. An alternative approach might involve uniformly suppressing

language priors, but recent analyses reveal that even for image-related tokens, the reliance on language priors increases as generation progresses, suggesting that such uniform suppression could degrade text quality and exacerbate hallucinations (Min et al., 2025). These observations indicate that effective hallucination mitigation should not aim to weaken language priors globally, but instead target visually descriptive tokens specifically.

Motivated by this insight, we design a reward mechanism that operates selectively on visually descriptive tokens. We define visually descriptive tokens as those that directly convey observable visual content, including objects, attributes (e.g., color and shape), spatial relations, and visually grounded actions, while excluding tokens primarily serving grammatical, connective, or abstract functions. To explicitly restrict optimization to such tokens, we assume access to a predefined set of visual-related token identifiers $\mathcal{V}_{\text{vis}} \subset \mathcal{V}$, and introduce a binary gating function for each generated token $y_t$:

$$g_t = \begin{cases} 1, & \text{if } y_t \in \mathcal{V}_{\text{vis}}, \\ 0, & \text{otherwise.} \end{cases} \qquad (5)$$

Rewards are applied only when $g_t = 1$, focusing optimization on tokens that require visual grounding while leaving non-visual tokens unconstrained. This selective design enhances visual fidelity without sacrificing linguistic fluency, and in contrast to decoding-time interventions, allows the model to internalize visual grounding constraints during training.

### 3.3. Hallucination-Aware Token Classification

Decoding-time analyses of vision-language models such as DeCo (Wang et al., 2024a) highlight a systematic layer-wise discrepancy between visually grounded tokens and hallucinated tokens during generation, which is further supported by recent studies on internal attention patterns (Jiang et al., 2025; He et al., 2025). Specifically, tokens that accurately describe visual content tend to receive strong probability support in intermediate layers, while their probabilities remain stable or slightly suppressed at the final layer. In contrast, hallucinated tokens often exhibit weak support in early and intermediate layers but are amplified in the final layers, where generation is increasingly influenced by linguistic priors. This phenomenon motivates us to treat the inconsistency between anchor and final layer probabilities as a key indicator for hallucination mitigation during training. Accordingly, we introduce a hallucination-aware token classification method that leverages internal model probability signals to distinguish hallucinated tokens from visually grounded ones. For a generated token $y_t$, we define its maximum anchor layer confidence as:

$$p_t^{\max} = \max_{l \in A} p_{t,y_t}^{(l)}, \qquad (6)$$

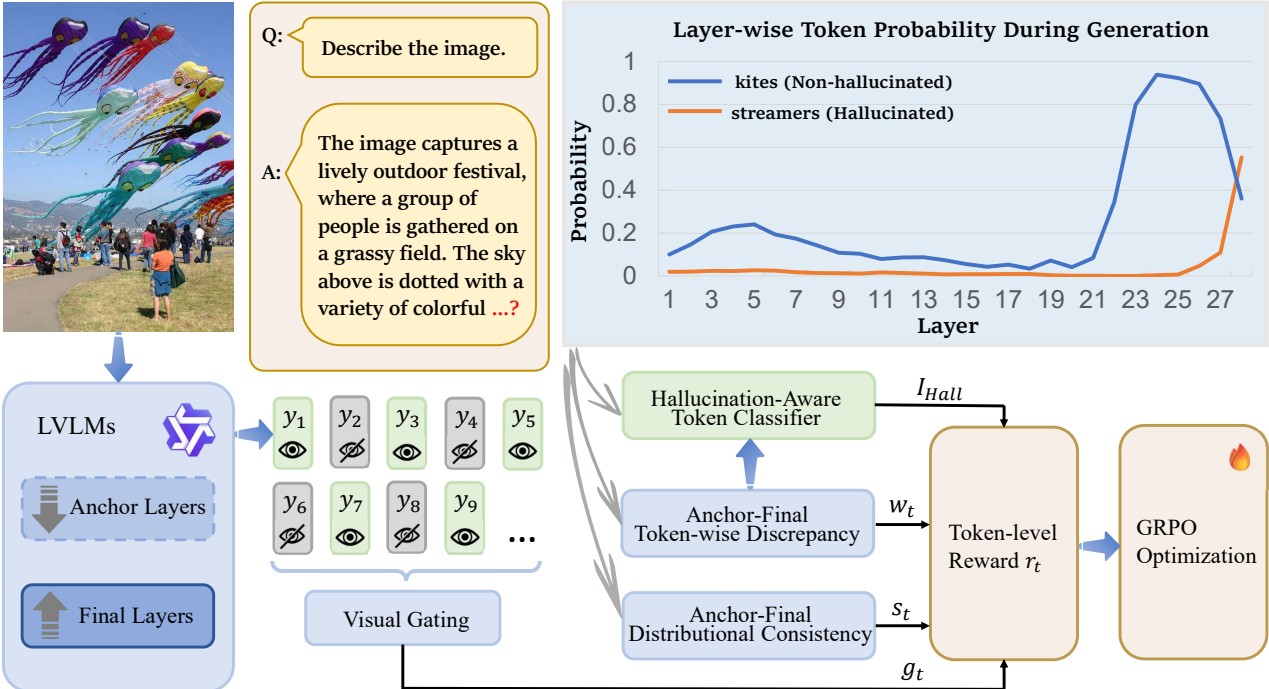

*Figure 2.* Overview of the AFS token-aware optimization framework. We first apply visual-selective gating to focus on visually descriptive tokens. For each gated token, we perform hallucination-aware token classification based on token-wise anchor-final probability discrepancies. We then construct a layer-wise anchor-final distributional consistency reward that penalizes late-stage amplification inconsistent with intermediate visual evidence, and optimize the LVLMs via GRPO using internal model signals.

where $A$ denotes the set of candidate anchor layers and $p_{t,y_t}^{(l)}$ is the probability assigned to $y_t$ at time step $t$ by layer $l$. We further define the anchor-final probability gap as:

$$\Delta_t = p_t^{\max} - p_{t,y_t}^{(L)}, \tag{7}$$

where $L$ is the index of the final layer. A negative gap indicates that the token is more weakly supported by anchor layers than by the final layer, suggesting that its probability is predominantly amplified in the later stages of generation. Therefore, we assume that a visual token is hallucinated if its anchor layer confidence falls within a moderate range and its probability is amplified at the final layer, formalized as:

$$I_{\text{hall}}(t) = \mathbb{I}\big(\varepsilon_1 < p_t^{\max} < \varepsilon_2 \ \wedge \ \Delta_t < 0\big), \tag{8}$$

where $\varepsilon_1$ is a small lower threshold close to zero, designed to filter out tokens with negligible support across all layers, and $\varepsilon_2$ is a higher confidence threshold, typically set around $0.4$, which excludes tokens that are already strongly supported by anchor layers and thus more likely to be visually grounded. Tokens that do not satisfy this condition are classified as non-hallucinated visual tokens:

$$I_{\text{non-hall}}(t) = 1 - I_{\text{hall}}(t). \tag{9}$$

Note that this classification does not aim to provide a ground-truth identification of hallucinations. Instead, it operationalizes a hypothesis inspired by DeCo (Wang et al., 2024a), where hallucinated visual tokens manifest insufficient intermediate-layer support and are amplified in later layers. This assumption is crucial because it allows the model to differentiate hallucinated tokens from visually grounded ones using internal signals, enabling effective training without requiring ground-truth labels or external supervision.

### 3.4. Anchor-Final Distributional Consistency Reward

Building upon visual-selective gating and hallucination-aware token classification, we define a unified token-level reward that simultaneously encourages visually grounded tokens and distributional consistency between anchor and final layer predictions. For each generated token $y_t$, we first identify the most confident anchor layer for this token as:

$$l^* = \arg\max_{l \in A} p_{t,y_t}^{(l)}, \tag{10}$$

we then define the corresponding probability distributions:

$$P_t = p_t^{(L)}, \quad Q_t = p_t^{(l^*)}, \tag{11}$$

where $P_t$ represents the predicted distribution at the final layer $L$, and $Q_t$ represents the predicted distribution at

the most confident anchor layer $l^*$. To quantify the alignment between intermediate visual evidence and the final prediction, we measure their discrepancy using the Jensen-Shannon divergence $\text{JS}(P_t \parallel Q_t)$, we define the distributional consistency score as

$$s_t = 1 - \text{JS}(P_t \parallel Q_t) \tag{12}$$

where larger values indicate stronger alignment between the anchor and final layer predictions. To capture the relative importance of each token, we define a positive token-specific weight as:

$$w_t = I_{\text{hall}}(t) \cdot (-\Delta_t) + I_{\text{non-hall}}(t) \cdot \max(\Delta_t, \varepsilon_3), \quad (13)$$

where $\varepsilon_3 > 0$ is a small constant ensuring that $w_t$ remains strictly positive and retains sufficient influence for non-hallucinated visual tokens even when the gap is small. Intuitively, $w_t$ reflects the reliability of intermediate-layer support: hallucinated tokens with strong final-layer amplification but weak anchor support receive larger weights via $-\Delta_t$, while visually grounded tokens are weighted by their positive anchor-final probability gap, with a lower bound to prevent vanishing contributions. The resulting visual-selective, hallucination-aware, anchor-final distributional consistency reward for token $y_t$ is defined as:

$$r_t = g_t \cdot \begin{cases} -\exp\left(\alpha(w_t \cdot s_t - c)\right), & \text{if } I_{\text{hall}}(t) = 1, \\ \exp\left(\alpha(w_t \cdot s_t - c)\right), & \text{otherwise,} \end{cases} \tag{14}$$

where $g_t$ is the visual gating indicator, $\alpha$ controls the sharpness of the reward, and $c$ is a constant offset.

From an optimization perspective, the proposed reward can be interpreted as an implicit regularizer that discourages late-stage probability amplification driven by language priors when such amplification is inconsistent with intermediate visual evidence. For hallucinated visual tokens, the reward is strictly negative, and its magnitude increases with both a larger token-wise anchor-final probability gap and higher layer-wise anchor-final distributional consistency. This design reflects the intuition that hallucinations are particularly detrimental when a token is amplified in the final layer, while the probability distribution of the entire vocabulary remains relatively consistent across layers, necessitating more aggressive penalties for such cases. In contrast, the reward is positive for non-hallucinated visual tokens. On the one hand, it increases with anchor-final distributional consistency, encouraging predictions that are consistently grounded in the visual evidence provided by intermediate layers. On the other hand, it is positively correlated with the token-wise probability gap. Specifically, when the gap is large, indicating that the token is suppressed in the final layer relative to its anchor layer confidence while the final prediction remains correct, the reward increases. This

highlights the value of weak signals, encouraging the model to correctly sample visually grounded tokens even when they are suppressed by competing linguistic priors at later stages, thus reinforcing the importance of low-probability but crucial tokens. Non-visual tokens are excluded from optimization via $g_t = 0$, ensuring that only visually descriptive tokens influence the model's learning and preventing any interference with the generation of grammatical, connective, or abstract tokens.

To obtain a sample-level reward for reinforcement learning, we aggregate token-level rewards within each generated sequence. For a response $y = (y_1, \ldots, y_T)$, we define:

$$R(y) = \frac{\sum_{t=1}^{T} r_t}{\sum_{t=1}^{T} g_t}. \tag{15}$$

This normalization ensures that the reward focuses on visually descriptive content and prevents sequences with more visually descriptive tokens from dominating the optimization.

In summary, by jointly applying visual-selective gating, hallucination-aware classification, and anchor-final distributional consistency, the proposed reward explicitly encourages the model to ground its predictions in visual evidence, mitigating hallucinations without relying on external supervision or ground-truth captions.

## 4. Experiments

### 4.1. Experimental Setup

**Baselines.** We compare our approach with several representative decoding-time hallucination mitigation methods, including DeCo (Wang et al., 2024a), VCD (Leng et al., 2024), ICD (Wang et al., 2024c) and DoLa (Chuang et al., 2023), as well as three training-time baselines, including MOCHa (Ben-Kish et al., 2024), HA-DPO (Zhao et al., 2023), and a standard GRPO baseline trained on GQA yes/no data with binary rewards. All baselines are implemented on Qwen2.5-VL-7B, following the hyperparameters in their original papers.

**Implementation Details.** We fine-tune Qwen2.5-VL-7B on a subset of MSCOCO 2014 (Lin et al., 2014) training split, consisting of 38,400 caption-free images. We optimize the model using GRPO (Shao et al., 2024) with the prompt "Describe the image." We employ a cosine learning rate scheduler, with a peak learning rate of $1 \times 10^{-5}$ and a warmup ratio of $0.02$. The global batch size is 16, and we sample 12 completions per input. We set the maximum completion length to 256 tokens and choose anchor layers $l \in \{20, 21, \ldots, 27\}$. The hallucination classification thresholds are set to $\varepsilon_1 = 0.01$ and $\varepsilon_2 = 0.4$, and the minimum weight for non-hallucinated tokens is set to $\varepsilon_3 = 0.1$.

Reward shaping is parameterized by $\alpha = 1.0$ and $c = 0.5$. At inference time, all methods use greedy decoding.

**Evaluation Metrics and Benchmarks.** We evaluate hallucination using CHAIR (Rohrbach et al., 2018), POPE (Li et al., 2023b), MME (Fu et al., 2025), AMBER (Wang et al., 2023) and MM-Vet (Yu et al., 2023). Details of each metric and benchmark are provided in Appendix A.

## 4.2. Main Results

**Hallucination Mitigation on Caption Generation.** We first evaluate hallucination mitigation on caption generation using CHAIR and the generative subset of AMBER. As shown in Table 1, our method significantly reduces object hallucinations, improving CHAIR$_S$ from 28.8 to 24.6 and CHAIR$_I$ from 9.1 to 7.7, outperforming all baselines. Notably, this reduction is accompanied by a modest increase in recall, from 64.3% to 64.6%, demonstrating that our method can suppress hallucinated visual tokens without sacrificing descriptive coverage and even slightly improves the recall of grounded content. On AMBER, our approach consistently improves generative hallucination metrics, reducing CHAIR from 5.2 to 4.5 and Hal from 28.2 to 24.5, while maintaining comparable coverage. This further validates that our hallucination-aware optimization improves visual faithfulness without degrading caption quality.

**Generalization to Discriminative Tasks.** Although our model is trained in a caption generation setting, it transfers effectively to discriminative hallucination and general multimodal benchmarks, as summarized in Table 2. On POPE, our method achieves the highest average performance, reaching 87.31% accuracy and 85.87% F1, reflecting more reliable object existence judgments. On MME, our approach improves both Perception and Cognition, yielding the best overall score of 2341.44. The improvement on Cognition also indicates that our hallucination-aware training enhances multimodal reasoning under visual constraints, rather than merely boosting low-level perception. On AMBER, our model improves discriminative accuracy from 82.0% to 82.8% and F1 from 87.6% to 88.2%, further demonstrating the generalization of learned grounding behavior beyond captioning. Notably, despite being trained on caption-free MSCOCO images, our method shows strong cross-dataset generalization on MME and AMBER, effectively transferring across diverse visual domains and avoiding overfitting to a specific dataset or captioning format.

Overall, our method consistently reduces hallucinations in caption generation while preserving descriptive coverage, and generalizes effectively to discriminative tasks and non-COCO datasets, demonstrating that our hallucination-aware optimization fosters robust and transferable multimodal understanding.

## 4.3. Analysis of Hallucination-Aware Token Classification

A central assumption of our approach is that hallucinated visual tokens manifest internal probability patterns that are fundamentally distinct from those of visually grounded tokens: they receive limited support from intermediate layers while being amplified at the final layer. We empirically validate this assumption by analyzing anchor-final probability distributions and evaluating the robustness of token classification across various confidence thresholds.

**Separation in Anchor-Final Probability Gap.** For each generated visual token $y_t$, we compute its maximum anchor-layer confidence $p_t^{\max}$, final-layer probability $p_{t,y_t}^{(L)}$, and the anchor-final probability gap $\Delta_t = p_t^{\max} - p_{t,y_t}^{(L)}$. Tokens are grouped along two orthogonal axes: (i) whether a token appears in the ground-truth caption, which serves as a weak indicator of its visual relevance, and (ii) whether it is classified as hallucinated or non-hallucinated by our hallucination-aware token classification. To assess the separation between these groups, we compute the Kullback-Leibler (KL) divergence between their anchor-final probability gap distributions, as shown in Table 3. Tokens classified as non-hallucinated under our method closely match those appearing in the caption, with a remarkably low KL divergence of 0.0004. In contrast, tokens classified as hallucinated show higher KL divergence, with their probability distributions aligning more closely with tokens absent from the caption than those present in it. This confirms that the anchor-final probability gap effectively differentiates visually grounded tokens from hallucinated ones, suggesting that our method provides a reliable self-supervision signal for hallucinated token identification.

*Table 3.* KL Divergence between hallucinated and non-hallucinated tokens under our classification, grouped by caption presence.

|  | In Caption | Not in Caption |
| --- | --- | --- |
| Non-hallucinated | 0.0004 | 0.1379 |
| Hallucinated | 0.6219 | 0.2397 |

**Sensitivity to Classification Thresholds.** Building on the observed separation in anchor-final probability gap, we evaluate the robustness of our hallucination-aware token classification with respect to the upper anchor-confidence threshold $\varepsilon_2$. Since ground-truth captions are weakly supervised and may not fully encompass all visually grounded tokens, we mitigate potential false positives by adjusting the prompt to generate more concise descriptions. We evaluate performance under different values of $\varepsilon_2$, reporting accuracy, precision, recall, and F1 score using ground-truth captions as weak supervision, as shown in Table 4. The results show that performance varies smoothly across a broad range of

*Table 1.* Results on generative hallucination benchmarks, including CHAIR and AMBER-Generative.

| Method | CHAIR | | | | AMBER-Generative | | | |
|---|---|---|---|---|---|---|---|---|
| | CHAIR$_S$ ↓ | CHAIR$_I$ ↓ | Recall↑ | Len | CHAIR↓ | Cover↑ | Hal.↓ | Cog.↓ |
| Vanilla | 28.8 | 9.1 | 64.3 | 109.5 | 5.2 | 64.3 | 28.2 | 1.7 |
| DeCo | 28.8 | 8.5 | 63.7 | 110.2 | 5.0 | 63.3 | 25.7 | 1.5 |
| VCD | 29.6 | 9.8 | 63.6 | 103.1 | 6.0 | 64.2 | 29.9 | 1.5 |
| ICD | 28.4 | 9.0 | 62.8 | 111.0 | 5.6 | 64.5 | 32.4 | 2.0 |
| DoLa | 30.0 | 9.0 | 63.9 | 115.2 | 6.9 | **66.5** | 39.2 | 2.9 |
| MOCHa | 25.6 | 8.3 | 61.9 | 95.3 | 4.7 | 63.9 | 25.1 | 1.6 |
| HA-DPO | 25.8 | **7.7** | 64.2 | 106.8 | 5.1 | 64.0 | 27.4 | 1.5 |
| Standard GRPO | 29.4 | 8.9 | **64.8** | 108.1 | 5.6 | 63.8 | 29.5 | 1.7 |
| **AFS (Ours)** | **24.6** | **7.7** | 64.6 | 106.7 | **4.5** | 63.9 | **24.5** | **1.4** |

*Table 2.* Results on discriminative benchmarks, including POPE, MME, and AMBER-Discriminative.

| Method | POPE | | MME | | | | AMBER-Discriminative | |
|---|---|---|---|---|---|---|---|---|
| | Accuracy↑ | F1-Score↑ | Hall↑ | Perception↑ | Cognition↑ | Total↑ | Accuracy↑ | F1-Score↑ |
| Vanilla | 86.29 | 84.49 | 705.00 | 1711.67 | 615.36 | 2327.03 | 82.0 | 87.6 |
| DeCo | 86.38 | 84.61 | **710.00** | 1716.67 | 613.21 | 2329.91 | 82.4 | 87.9 |
| VCD | 87.16 | 85.73 | 683.33 | 1677.00 | 600.71 | 2277.71 | 81.0 | 86.8 |
| ICD | 85.86 | 83.86 | 688.33 | 1665.62 | 582.86 | 2248.48 | 80.4 | 86.6 |
| DoLa | 81.47 | 77.38 | 638.33 | 1442.00 | 502.86 | 1944.86 | 77.7 | 85.4 |
| MOCHa | 86.80 | 85.14 | 705.00 | 1696.98 | 616.07 | 2313.05 | 82.5 | 88.0 |
| HA-DPO | 87.12 | 85.32 | 700.00 | 1710.29 | 596.07 | 2306.36 | 82.4 | 87.9 |
| Standard GRPO | **87.81** | **86.28** | **710.00** | 1706.51 | **625.71** | 2332.22 | 82.4 | 88.0 |
| **AFS (Ours)** | 87.31 | 85.87 | **710.00** | **1720.01** | 621.43 | **2341.44** | **82.8** | **88.2** |

thresholds, with F1 scores remaining stable and peaking at a moderate value of $\varepsilon_2 = 0.4$. This stability suggests that our classification method is robust to threshold selection.

*Table 4.* Sensitivity of hallucination-aware token classification to the upper anchor-confidence threshold $\varepsilon_2$.

| $\varepsilon_2$ | Precision | Recall | Accuracy | F1-Score |
|---|---|---|---|---|
| 0.3 | 72.36 | 73.14 | 96.39 | 83.17 |
| **0.4** | **74.29** | **75.21** | **96.42** | **84.50** |
| 0.5 | 71.88 | 72.66 | 96.23 | 82.80 |
| 0.6 | 71.44 | 72.30 | 96.01 | 82.49 |

These analyses confirm that the anchor-final probability gap provides a reliable internal cue for identifying hallucinated visual tokens under our classification method. The observed alignment and separation between token groups, along with the method's robustness to threshold selection, validate the use of this gap as an effective self-supervision signal for hallucination-aware optimization.

### 4.4. Ablation Studies

**Ablation on Reward Factors.** Our reward function consists of four separable components, corresponding to visual-selective gating, hallucination-aware token classification, anchor-final distributional consistency, and token-specific weight based on the token-wise probability gap. Although

conceptually organized into three modules in Section 3, these factors can be independently disabled for a detailed ablation analysis. We consider the following variants:

- **w/o visual-selective gating (w/o $g_t$):** remove visual gating by setting $g_t = 1$, applying the reward uniformly across both visual and non-visual tokens.

- **w/o hallucination-aware token classification (w/o $I_{hall}$):** treat all visually descriptive tokens as non-hallucinated and assign them a non-hallucinated token-level reward, regardless of their token-wise probability gap.

- **w/o anchor-final distributional consistency (w/o $s_t$):** remove the alignment between anchor and final layer predictions by setting $s_t = 1$ for all tokens.

- **w/o token-specific weight (w/o $w_t$):** set $w_t = 1$ for all tokens, disregarding the token-specific weight derived from the token-wise probability gap.

As shown in Table 5, all variants outperform the vanilla model, with the complete reward formulation achieving the best performance. This pattern highlights the positive contribution of each reward factor to hallucination mitigation, while their combination offers synergistic optimization signals, yielding the best trade-off between hallucination suppression, content coverage, and output conciseness.

Table 6. Ablation study of anchor layers selection.

| Anchor Layers | CHAIR$_S$ ↓ | CHAIR$_I$ ↓ | Recall↑ | Len |
|---|---|---|---|---|
| Vanilla | 28.8 | 9.1 | 64.3 | 109.5 |
| 14–21 | 25.8 | 8.0 | **65.2** | 109.7 |
| 17–24 | 25.6 | 7.8 | 64.7 | 109.0 |
| **20–27** | **24.6** | **7.7** | 64.6 | 106.7 |

Table 7. Ablation study of reward sharpness $\alpha$.

| $\alpha$ | CHAIR$_S$ ↓ | CHAIR$_I$ ↓ | Recall↑ | Len |
|---|---|---|---|---|
| Vanilla | 28.8 | 9.1 | 64.3 | 109.5 |
| 0.5 | 25.0 | 8.1 | **65.3** | 109.2 |
| **1.0** | **24.6** | **7.7** | 64.6 | 106.7 |
| 2.0 | 25.4 | 8.3 | 65.1 | 109.7 |

Table 5. Ablation study of reward factors.

| Variant | CHAIR$_S$ ↓ | CHAIR$_I$ ↓ | Recall↑ | Len |
|---|---|---|---|---|
| Vanilla | 28.8 | 9.1 | 64.3 | 109.5 |
| w/o $g_t$ | 26.2 | 8.0 | 65.5 | 108.1 |
| w/o $I_{\text{hall}}$ | 27.2 | 8.8 | **65.7** | 111.9 |
| w/o $s_t$ | 25.6 | 8.0 | 63.9 | 109.6 |
| w/o $w_t$ | 25.8 | 8.3 | 64.5 | 108.4 |
| **AFS** | **24.6** | **7.7** | 64.6 | 106.7 |

Among all ablations, removing hallucination-aware token classification ($I_{\text{hall}}$) leads to the most pronounced degradation of hallucination metrics, with CHAIR$_S$ and CHAIR$_I$ increasing to 27.2 and 8.8 respectively, despite achieving the highest recall. This pattern reveals an implicit optimization bias: when hallucinated tokens are not explicitly distinguished, the model favors coverage and linguistic fluency, which allows visually descriptive tokens with weak visual support to be amplified. Our hallucination-aware classification serves as a critical mechanism to address this bias by detecting such tokens through anchor-final probability discrepancies and assigning corrective rewards, thereby suppressing amplification driven by language priors and favoring tokens supported by reliable visual evidence.

**Ablation on Anchor Layers Selection.** Table 6 studies the sensitivity of our method to the anchor layers selection. Across various anchor configurations, all variants significantly outperform the vanilla model, suggesting that our proposed reward is not overly sensitive to the specific choice of anchor layers. Anchors selected from mid-to-late layers consistently achieve lower hallucination rates and slightly shorter outputs, indicating that representations at these stages provide more informative visual evidence for effective hallucination suppression without compromising recall.

**Ablation on Reward Sharpness $\alpha$.** Table 7 evaluates the sensitivity of our method to the reward sharpness parameter $\alpha$. Across the tested values, all settings consistently outperform the vanilla baseline, indicating low sensitivity to precise tuning. Moderate sharpness at $\alpha = 1.0$ yields slightly stronger hallucination suppression with shorter outputs, while both smaller and larger values remain effective, demonstrating that the proposed reward provides a robust and effective optimization signal.

## 5. Related Work

**Internal Model Signals for Hallucination Mitigation in Large Vision-Language Models.** Hallucinations in LVLMs often arise from the imbalance between visual evidence and language priors. Recent approaches utilize internal model signals, such as attention patterns, layer-wise representations, and latent space features, to address this issue. OPERA (Huang et al., 2024) reduces hallucinations by reallocating attention and penalizing over-trusted tokens in self-attention, while (Jiang et al., 2025) enhances visual grounding by adjusting attention to tokens with weaker visual support in intermediate layers. Similarly, (Yang et al.) targets hallucination heads within multi-head attention modules and reduces their reliance on text tokens, and TARAC (Xie et al., 2025), VHR (He et al., 2025) enhance visual attention during inference and refine vision-aware heads. VTI (Liu et al., 2024b) and (Liu et al., 2025) apply latent-space steering to stabilize visual features and improve alignment, while (Huo et al., 2024) further refines the visual-textual alignment by using introspection. Other methods, like (Ferrando et al., 2024), leverage sparse autoencoders to identify hallucination-related representations, and (Hu et al., 2025) addresses hallucinations from a Bayesian perspective through prior rectification, redundant visual token removal, and early stopping. In contrast to these methods, our approach leverages internal model signals as a training-time objective, enabling the model to internalize visual grounding constraints and mitigate hallucinations during training without post-hoc corrections or inference-time interventions.

**Training-Time Hallucination Mitigation.** Hallucination in LVLMs has been actively addressed through training-time interventions that aim to improve visual grounding by refining training data or optimization objectives. A prominent line of work adopts data-centric strategies, such as constructing large-scale instruction data with both positive and negative samples (Liu et al., 2023; Yu et al., 2024a; Chen et al., 2025), or incorporating fine-grained annotations to encourage more grounded reasoning during training (Gunjal et al., 2024; Zhang et al., 2024a). Although effective, these methods rely on large-scale curated datasets or detailed annotations. Another line of work mitigates hallucination through training-time alignment and preference optimization. MOCHa (Ben-Kish et al., 2024) formulates hallucination as a sequence-level optimization problem with multi-objective rewards, while methods like Factually Aug-

mented RLHF (Sun et al., 2024), RLHF-V (Yu et al., 2024b), HA-DPO (Zhao et al., 2023), Silkie (Li et al., 2023a), and POVID (Zhou et al., 2024b) further frame hallucination as an alignment issue and reduce it via preference optimization. Despite their success, these approaches typically depend on explicit supervision, reward models, human or AI-generated feedback. In contrast, our method mitigates hallucination at training time using internal model signals, allowing the model to internalize visual grounding constraints without external supervision or feedback.

## 6. Conclusion

In this paper, we propose AFS: Anchor-Final Self-Supervision, a novel self-supervised framework for mitigating hallucinations in large vision-language models. Motivated by the anchor-final amplification pattern on hallucinated tokens, AFS employs visual-selective rewards and distinguishes hallucinated tokens from visually grounded ones based on the token-wise probability gap, while enforcing consistency between intermediate and final layer probability distributions. Extensive experiments across multiple benchmarks demonstrate that our method significantly reduces hallucinations on generative tasks, generalizes well to discriminative tasks, and transfers effectively across diverse visual domains. These results highlight the effectiveness and generalizability of AFS as a scalable approach for hallucination-aware training in LVLMs.

## Acknowledgements

Thanks to the reviewers for their constructive comments and valuable suggestions. This work is supported by New Generation Artificial Intelligence-National Science and Technology Major Project (No. 2025ZD0122901). This work is also supported by the National Natural Science Foundation of China (No. 62572313, No. 62106139).

## Impact Statement

This paper improves the reliability of large vision-language models by reducing visual hallucinations, making generated outputs more faithful to visual evidence. This improvement can significantly benefit tasks that require reasoning and multimodal understanding, such as visual question answering and content verification, where incorrect visual descriptions may lead to misleading conclusions. However, improved reliability may also increase user confidence in model outputs and could be misused to produce persuasive misinformation or manipulated content. Overall, we expect this work to contribute to safer and more reliable multimodal AI, with deployment in reasoning tasks requiring careful risk assessment and ongoing monitoring to prevent unintended consequences.

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

# A. Evaluation Metrics and Benchmarks

## A.1. CHAIR

We adopt the Caption Hallucination Assessment with Image Relevance (CHAIR) (Rohrbach et al., 2018) metric to quantify object-level hallucination in generated descriptions. CHAIR measures the proportion of mentioned objects that are not grounded in the image, based on ground-truth captions and object segmentations. It evaluates hallucinations at both the instance level ($\text{CHAIR}_\text{I}$) and the sentence level ($\text{CHAIR}_\text{S}$), defined as:

$$\text{CHAIR}_\text{I} = \frac{|\{\text{hallucinated objects}\}|}{|\{\text{all objects mentioned}\}|}, \tag{16}$$

$$\text{CHAIR}_\text{S} = \frac{|\{\text{descriptions with hallucinated objects}\}|}{|\{\text{all descriptions}\}|}. \tag{17}$$

We evaluate on a subset of 500 images from the MSCOCO 2014 validation split using the prompt "Describe the image."

## A.2. POPE

The Polling-based Object Probing Evaluation (POPE) (Li et al., 2023b) is a VQA-style benchmark for evaluating object hallucination via binary visual questions of the form *"Is there a/an <object> in the image?"*. POPE adopts three object sampling strategies with increasing difficulty: *random*, *popular*, and *adversarial*, which probe nonexistent objects that are respectively randomly selected, frequently occurring, or contextually plausible given the image. We conduct evaluation on the MSCOCO subset and report F1 score and accuracy.

## A.3. MME

The Multimodal Large Language Model Evaluation benchmark (MME) (Fu et al., 2025) is a comprehensive evaluation suite designed to assess the general capabilities of large vision-language models across a diverse set of tasks. MME consists of 10 perception tasks and 4 cognition tasks, covering both visual understanding and multimodal reasoning abilities. Each task is evaluated independently, and task-specific scores are aggregated to produce overall Perception and Cognition scores, which are reported as the primary evaluation metrics.

## A.4. AMBER

AMBER (Wang et al., 2023) is a multi-dimensional benchmark for evaluating hallucinations in LVLMs across generative and discriminative tasks. It measures existence, attribute, relation, and hallucinatory target objects. Generative evaluation uses CHAIR, Cover, Hal, and Cog, while discriminative evaluation relies on binary visual queries scored with Accuracy, Precision, Recall, and F1. The AMBER Score combines generative CHAIR and discriminative F1 into a single metric. All evaluations are performed on 1,004 curated images.

## A.5. MM-Vet

MM-Vet (Yu et al., 2023) evaluates LVLMs on complex tasks requiring six core vision-language capabilities: Recognition, Knowledge, OCR, Spatial Awareness, Language Generation, and Math, including their 16 integrations. It contains 218 open-ended questions covering diverse real-world scenarios. Evaluation is performed with an LLM-based scorer using few-shot examples of correct, incorrect, and partially correct answers, producing a 0–1 score per sample.

# B. Implementation Details

## B.1. Construction of Visually Descriptive Token Set

Our method relies on a predefined set of visually descriptive tokens $\mathcal{V}_\text{vis}$ to enable visual-selective reward gating. We adopt an automatic, model-assisted token classification procedure to classify tokens into visually descriptive and non-visual categories, which is constructed offline prior to training, generated only once and can be permanently reused across multiple experiments. Specifically, each token in the tokenizer vocabulary is independently classified by a large vision-language model using a natural language prompt. The model is instructed to determine whether a token conveys observable visual content, such as objects, attributes (e.g., color and shape), spatial relations, or visually grounded actions, or instead serves

primarily grammatical, connective, abstract, or functional roles. Formally, for each token $t$, we query the model with the following instruction:

```
You classify tokens into VISUAL or NON_VISUAL. VISUAL tokens:  objects,
attributes, colors, shapes, relations, positions, visually observable
actions.  NON_VISUAL tokens:  stopwords, grammar words, connectors,
abstract terms, emotions, punctuation.  Only output one word:  VISUAL or
NON_VISUAL.
```

The model's response is parsed to assign a binary label to each token. Tokens labeled as VISUAL are included in $\mathcal{V}_{\text{vis}}$, while all others are treated as non-visual tokens. Note that the resulting set of visual tokens represents a minority of the overall vocabulary, ensuring that optimization selectively targets visually descriptive tokens without interfering with general language modeling. We use Qwen2.5-VL-7B to classify the vocabulary, which is sufficiently powerful to handle the task of identifying visually descriptive tokens with high reliability. In summary, this predefined set of visual tokens remains accurate and stable, requiring no frequent updates or manual corrections.

### B.2. POPE Evaluation Prompt

For evaluation on the POPE, we follow a standardized binary question-answering protocol. POPE consists of binary visual questions of the form *"Is there a/an <object> in the image?"*, designed to probe object-level hallucination. Each question is converted into a yes/no format using the following prompt template:

```
Please answer the following question with yes or no.  Question:
<question_text> Answer:
```

All methods are evaluated with the same prompt and greedy decoding to ensure a fair comparison. The model's generated answer is post-processed to extract a binary decision, which is then used to compute accuracy and F1 scores following the POPE evaluation protocol.

## C. Detailed Results on POPE

We present the detailed evaluation results of the POPE benchmark in Table 8, where our method outperforms all comparison methods across all categories, including the Adversarial, Popular, and Random subsets. These results highlight the effectiveness of our approach in enhancing the model's performance on discriminative tasks.

*Table 8.* Detailed POPE evaluation results.

| Method | Adversarial | | Popular | | Random | | Average | |
|---|---|---|---|---|---|---|---|---|
| | Accuracy↑ | F1 score↑ | Accuracy↑ | F1 score↑ | Accuracy↑ | F1 score↑ | Accuracy↑ | F1 score↑ |
| Vanilla | 85.60 | 83.83 | 86.23 | 84.43 | 87.03 | 85.20 | 86.29 | 84.49 |
| DeCo | 85.70 | 83.96 | 86.30 | 84.53 | 87.13 | 85.33 | 86.38 | 84.61 |
| VCD | 86.43 | 85.05 | 86.93 | 85.50 | 88.10 | 86.63 | 87.16 | 85.73 |
| ICD | 85.33 | 83.36 | 85.80 | 83.80 | 86.43 | 84.41 | 85.86 | 83.86 |
| DoLa | 81.33 | 77.25 | 81.40 | 77.32 | 81.67 | 77.57 | 81.47 | 77.38 |
| **AFS (Ours)** | **86.53** | **85.13** | **87.23** | **85.79** | **88.17** | **86.69** | **87.31** | **85.87** |

## D. Detailed Results on AMBER Discriminative tasks

We present the detailed results of the AMBER discriminative tasks in Table 9, showing that our method outperforms the vanilla model across most categories, including Existence, Attribute, Number, Action, and notably Relation category. These improvements highlight the effectiveness of our approach in advancing performance across a range of multimodal reasoning tasks.

*Table 9.* Performance comparison on AMBER discriminative tasks.

| Methods | Existence | | Attribute | | State | | Number | | Action | | Relation | | Total | | AMBER Score |
|---|---|---|---|---|---|---|---|---|---|---|---|---|---|---|---|
| | Acc | F1 | Acc | F1 | Acc | F1 | Acc | F1 | Acc | F1 | Acc | F1 | Acc | F1 | |
| Vanilla | 95.0 | 97.4 | **78.5** | 81.8 | **75.0** | **79.3** | 86.5 | 87.9 | 78.5 | 82.1 | 60.0 | 67.4 | 82.0 | 87.6 | 91.20 |
| **AFS (Ours)** | **95.9** | **97.9** | **78.5** | **81.9** | 74.2 | 78.9 | **88.1** | **89.3** | **79.2** | **82.6** | **63.3** | **69.1** | **82.8** | **88.2** | **91.85** |

## E. Results on MM-Vet Benchmark

We evaluate our method on the MM-Vet benchmark using Qwen3-8B as the scoring model. The results summarized in Table 10 demonstrate that our method outperforms the vanilla model across several key metrics, achieving improvements in Recognition, OCR, and particularly in Spatial awareness and Math. These results suggest that our method effectively enhances reasoning capabilities on complicated multimodal tasks, and further demonstrates strong cross-dataset generalization.

*Table 10.* Performance comparison on the MM-Vet benchmark.

| Method | Rec ↑ | OCR ↑ | Know ↑ | Gen ↑ | Spat ↑ | Math ↑ | Total ↑ |
|---|---|---|---|---|---|---|---|
| Vanilla | 52.6 | 59.9 | **50.0** | **53.7** | 54.8 | 53.8 | 54.5 |
| **AFS (Ours)** | **54.3** | **61.4** | 49.2 | 53.1 | **56.8** | **57.7** | **56.5** |

## F. Limitation

One limitation of our method is that although it is conceptually model-agnostic, it relies on access to intermediate-layer probabilities and is therefore primarily applicable to open-source LVLMs or architectures that expose internal states, and it may not directly extend to closed-source APIs or systems where such representations are unavailable.

In addition, the supervision signal is derived from the model's own internal distributions. Although our experiments across multiple backbones demonstrate consistent improvements, this design may still inherit or amplify systematic blind spots when the underlying visual evidence is consistently misinterpreted, and further validation on a broader range of large-scale or proprietary models would be valuable to fully assess the practical applicability of the proposed framework.

## G. Empirical Robustness to the Hallucination Classification Threshold $\epsilon_2$

The hallucination-aware token classifier employs the upper anchor-confidence threshold $\epsilon_2$ to exclude tokens that already receive strong support from anchor layers. To assess the sensitivity of AFS to this threshold, we evaluate its performance across a range of $\epsilon_2$ values under two additional settings: training on a different dataset Flickr30k and using a different backbone InternVL2.5-4B. Specifically, we vary $\epsilon_2$ over $\{0.3, 0.4, 0.5, 0.6\}$ and evaluate hallucination mitigation using CHAIR metrics and recall.

*Table 11.* Robustness of AFS to the upper anchor-confidence threshold $\epsilon_2$ across a different training dataset and model architecture.

| Setting | Flickr30k | | | InternVL2.5-4B | | |
|---|---|---|---|---|---|---|
| | CHAIR$_S$ ↓ | CHAIR$_I$ ↓ | Recall↑ | CHAIR$_S$ ↓ | CHAIR$_I$ ↓ | Recall↑ |
| Vanilla | 28.8 | 9.1 | 64.3 | 38.0 | 8.0 | 70.5 |
| $\epsilon_2 = 0.3$ | 25.4 | 7.8 | 64.2 | 33.6 | 7.8 | 69.4 |
| $\epsilon_2 = 0.4$ | **25.2** | **7.7** | **64.9** | 33.4 | 7.5 | 70.3 |
| $\epsilon_2 = 0.5$ | **25.2** | 8.2 | 64.5 | **32.2** | **7.4** | 70.4 |
| $\epsilon_2 = 0.6$ | 25.8 | 8.3 | 64.8 | 32.8 | **7.4** | **70.6** |

As shown in Table 11, AFS consistently reduces hallucination across all tested values of $\epsilon_2$. To further explain this stability, we analyze the distribution of the maximum anchor-layer confidence $p_t^{\max}$. As shown in Table 12, most tokens lie in either low-confidence or high-confidence regions, while only 14.55% fall into the intermediate range $[0.3, 0.6)$ where changes in $\epsilon_2$ directly affect the hallucination-aware classification. This concentration pattern indicates that changing $\epsilon_2$ mainly affects

a small set of tokens and has limited impact on the overall reward assignment. Overall, the results demonstrate that the hallucination-aware token classifier is robust to the choice of $\epsilon_2$, and that the gains of AFS remain stable across both training dataset and model architecture changes.

*Table 12.* Distribution of maximum anchor-layer confidence $p_t^{\max}$.

| $p_t^{\max}$ | Proportion |
|---|---|
| $[0.0, 0.3)$ | 36.64% |
| $[0.3, 0.6)$ | 14.55% |
| $[0.6, 1.0)$ | 48.81% |

## H. Computational Cost and Efficiency

We analyze the computational overhead of AFS relative to training-time baselines, including SFT and standard GRPO. As shown in Table 13, the dominant cost in our framework comes from GRPO, which requires sampling multiple rollouts ($K = 12$ by default). Compared to standard GRPO, AFS introduces only modest overhead, with a 7.8% increase in wall-clock time and a 3.4% increase in peak memory. This additional cost comes from exposing intermediate-layer representations during rollout and computing layer-wise probabilities for reward construction. Notably, the reward computation is gradient-free and does not introduce additional backpropagation or auxiliary networks.

*Table 13.* Computational cost comparison.

| Method | Time / Step (s) | Peak Memory (MB) |
|---|---|---|
| SFT | 1.34 | 51,355 |
| Standard GRPO | 24.75 | 126,993 |
| AFS (Ours) | 26.67 | 131,349 |

Therefore, compared to decoding-time methods, AFS incurs cost only during training and has no inference overhead. Meanwhile, compared to GRPO-based methods, the additional cost of AFS is minor relative to rollout sampling. Overall, AFS achieves a favorable trade-off between efficiency and effectiveness, with most of the overhead coming from GRPO itself rather than our proposed components.

