# OpenReview forum: "Anchor-Final Self-Supervision Drives Hallucination-Aware Optimization in Large Vision-Language Models"
_ICML.cc/2026/Conference — ICML 2026 regular_

### Official Review · Reviewer_caJo · 2026-03-10

**Soundness:** 2
**Presentation:** 2
**Significance:** 3
**Originality:** 3
**Overall Recommendation:** 4
**Confidence:** 3

**Summary:**

This paper proposes AFS (Anchor-Final Self-Supervision), a training time hallucination mitigation method for LVLMs that uses only model internal layer-wise prediction signals at optimization time. The core idea is that hallucinated visual tokens tend to receive weaker support in intermediate “anchor” layers and stronger support at the final layer, reflecting late amplification by language priors. Based on this intuition, the method combines three components: visual-selective reward gating over “visual” tokens, hallucination-aware token classification using anchor-final probability gaps, and a reward encouraging anchor-final distributional consistency. The model is then fine-tuned with GRPO on caption-free MSCOCO images. Empirically, the paper reports improved hallucination metrics on CHAIR and AMBER-Generative, competitive or best performance on POPE/MME/AMBER-Discriminative, robustness across several ablations, and modest gains on MM-Vet.

**Compliance With Llm Reviewing Policy:**

Affirmed.

**Final Justification:**

As authors mitigated my concerns. update my final decision.

**Key Questions For Authors:**

1. Why are there no comparisons against training-time hallucination mitigation baselines? Since AFS is itself a training-time method, comparing only to decoding-time methods makes the positioning incomplete. A direct comparison to at least one representative training-time baseline would materially affect my evaluation.
2. How robust is AFS beyond the single training prompt “Describe the image.”? The current training/evaluation setup is narrow. If you can show transfer to more diverse instruction-following prompts or conversational prompts, that would strengthen both significance and soundness.
3. How sensitive is the method to the choice of weak supervision and thresholds used in hallucination-aware classification? The paper studies $\epsilon_2$, but I would like more evidence on the stability of the classifier assumptions more broadly, especially since caption presence is only a weak proxy for visual grounding. A stronger analysis here could increase my confidence in the mechanism.
4. What is the computational cost relative to decoding-time baselines and to other training-time alternatives? Since AFS requires GRPO fine-tuning with multiple completions per image, training cost is relevant to practical impact. If the compute is reasonable or especially sample-efficient, that would improve my view of significance.

**Limitations:**

No. The paper has an impact statement and a brief appendix limitation noting that experiments are limited to the Qwen2.5-VL backbone, but the discussion is not broad enough. It should also discuss dependence on the externally constructed visual-token set, the heuristic nature of the hallucination classifier, the lack of training-time baseline comparisons, the narrow training prompt distribution, and computational cost.

**Strengths And Weaknesses:**

Strengths: Hallucination mitigation in LVLMs is an important problem, and the paper tackles it from a training-time angle rather than only at decoding time. The fact that the method is trained on caption-free images and still transfers to POPE, MME, AMBER, and MM-Vet is a meaningful practical point. The idea of turning layerwise internal disagreement into a self-supervised optimization signal is potentially useful beyond this exact formulation. Additionally, the more interesting contribution is the specific anchor-final view of hallucinations and how it is operationalized into token gating, token classification, and a consistency reward. That is a reasonably original way to turn internal model dynamics into a training signal, even though GRPO and layer-contrast intuitions are themselves not new.

Weaknesses: My main concern is that the central hallucination signal remains largely heuristic. The paper assumes that hallucinated tokens are characterized by moderate anchor confidence and negative anchor-final gap, but this remains an empirically motivated proxy rather than a strongly validated mechanism. The supporting analysis is suggestive, but it still relies on weak supervision from caption presence and a threshold sweep over a single parameter, which is not enough to establish that the classifier reliably captures hallucinations in a broader sense.  A second soundness issue is that the “solely internal model signals” claim is overstated. The appendix shows that the visual-token set V_vis is constructed offline by querying Qwen2.5-VL-7B to label each vocabulary token as VISUAL or NON VISUAL. That is a model-assisted external preprocessing step, not purely an internal signal extracted from the trained policy during optimization. This does not invalidate the method, but it weakens one of the paper’s main framing claims and should be described more carefully. A third issue is experimental positioning. The paper presents AFS as a training-time hallucination mitigation method, but the experimental baselines are only decoding-time methods (DeCo, VCD, ICD, DoLa). The related-work section itself lists several training-time approaches, including data-centric and preference/alignment-based methods, yet there is no direct empirical comparison against those classes of methods. That makes it difficult to judge whether the gains come from the anchor-final idea specifically or simply from allowing any training-time adaptation. Moreover, there are also some scope limitations in the evaluation. Training uses a single captioning-style prompt (“Describe the image.”) on 38,400 caption-free MSCOCO images with greedy decoding at inference. This is a fairly narrow setting relative to the broader LVLM usage scenarios invoked in the paper, and it leaves open how well the method transfers to more instruction-following or conversational prompts.

---

> ### Author Rebuttal · Authors · 2026-03-30
>
> **Q1 and W3: Comparison with Training-Time Baseline**
>
> We agree that comparisons with training-time baselines are essential for properly positioning our method. Our initial experiments did not include such baselines due to potential unfairness arising from differences in training data, task formulation, and backbone, which are absent in decoding-time comparisons. Following suggestions provided by you and Reviewer 2Si6, we identify MOCHa[1] as the closest training-time baseline as it is a captioning-based RL training method consistent with our setting. We further include a standard GRPO baseline (trained on GQA yes/no data with binary rewards) to isolate the effect of training-time optimization, and a representative preference-based method HA-DPO[2] for broader comparison. We hope this comparison better clarifies the positioning of our method, and shows that the gains are not merely due to generic training-time adaptation, but are closely related to optimizing the anchor–final signal.
>
> | Method        | CHAIR_S ↓ | CHAIR_I ↓ | Recall ↑ | Len   | POPE-F1 ↑    | MME ↑       | AMBER-Score ↑ | MM-Vet ↑ |
> | ------------- | --------- | --------- | -------- | ----- | ------------ | ----------- | ------------- | -------- |
> | Vanilla       | 28.8      | 9.1       | 64.3     | 109.5 | 84.49        | 2327.03     | 91.20         | 54.5     |
> | MOCHa         | 25.6      | 8.3       | 61.9     | 95.3  | 85.14        | 2313.05     | 91.65         | 55.9     |
> | HA-DPO        | 25.8      | **7.7**   | 64.2     | 106.8 | 85.32        | 2306.36     | 91.40         | 53.9     |
> | Standard GRPO | 29.4      | 8.9       | 64.8     | 108.1 | **86.28**    | 2332.22     | 91.20         | 55.2     |
> | AFS (Ours)    | **24.6**  | **7.7**   | 64.6     | 106.7 | $\underline{\text{85.87}}$ | **2341.44** | **91.85**     | **56.5** |
>
> [1]. Mitigating Open-Vocabulary Caption Hallucinations.
>
> [2]. Beyond Hallucinations: Enhancing LVLMs through Hallucination-Aware Direct Preference Optimization.
>
>
>
> **Q2 and W4: Robustness Beyond Single-Prompt Training**
>
> Thank you for raising this important question. Although training uses a single captioning prompt, our evaluation covers multiple instruction prompts at inference (e.g., POPE, MME, AMBER, MM-Vet). We observe consistent improvements across all settings, which demonstrates the generalization ability of our approach and its robustness beyond single-prompt training.
>
>
>
>
>
>
>
> **Q3 and W1: On the Hallucination-Aware** **Classifier** **as an Empirical but** **Robust** **Training Signal**
>
> Thank you for highlighting this main concern. We agree that the classifier is an empirically motivated signal rather than a fully validated detector of hallucination. However, our goal is not to explicitly define hallucination, but to construct a stable and informative training signal that correlates with hallucination tendencies and effectively guide optimization, without requiring precise classification. For further evidence on stability, please refer to our response to Reviewer 1vN2 (W1 and Q1: Robustness of Threshold $\epsilon_2$ and Training Stability), where we provide additional analysis demonstrating that the performance improvements are stable across datasets and model architectures.
>
>
>
> **W2: More Careful Description of the “Internal Signals” Claim**
>
> Thank you for pointing out this issue. We agree that “solely internal model signals” is too strong, and a more careful description is that our method primarily relies on internal signals for optimization without requiring annotated training data.
>
>
>
> **Q4: Computational Cost**
>
> Thank you for this important question. The additional overhead is modest (less than 8%) relative to standard GRPO and incurs no inference-time cost. Detailed analysis is provided in our response to Reviewer Mt9R (W2: Computational Cost and Efficiency), and we hope this clarifies that our method remains computationally practical.

---

> > ### Author Rebuttal · Reviewer_caJo · 2026-04-03
> >
> > Thanks to the author's effort. My concerns have been adequately addressed.

---

> > > ### Author Response · Authors · 2026-04-03
> > >
> > > Thanks to the reviewer for acknowledging that our responses have addressed your concerns. We greatly appreciate your insightful comments and the effort spent on reviewing our work.

---

### Official Review · Reviewer_1vN2 · 2026-03-11

**Soundness:** 3
**Presentation:** 2
**Significance:** 3
**Originality:** 3
**Overall Recommendation:** 4
**Confidence:** 4

**Summary:**

This paper introduces Anchor-Final Self-Supervision (AFS), a novel training framework designed to mitigate hallucinations in Large Vision-Language Models (LVLMs) without requiring external labels or human-in-the-loop feedback. The core technical insight is to leverage internal model signals by identifying the discrepancy between intermediate ('anchor') layer representations and final layer predictions.

Informed by prior work, 'hallucinated' or visually unsupported tokens often exhibit minimal support in early-to-middle layers but become dominant in the final output due to over-reliance on language priors. By supervising the model to align these final predictions more closely with the visually-grounded anchor layers, AFS effectively penalizes the generation of unsupported content. Empirical experiments demonstrate that this self-supervised alignment improves the truthfulness of LVLM outputs across several multimodal benchmarks.

**Compliance With Llm Reviewing Policy:**

Affirmed.

**Final Justification:**

The authors addressed my concerns in the rebuttal. I hence maintain my positive recommendation, and urge author to update the camera-ready to improve the clarity and quality.

**Key Questions For Authors:**

1. Hyperparameter Robustness (ϵ2) and Training Stability

- The robustness of the threshold ϵ2 is currently supported by a relatively restricted experimental view. While metrics appear to peak at 0.4, it is unclear if this value remains optimal across different model architectures, datasets, or other tasks.

- Given that the hallucination-aware token classification loss is a primary driver for metric improvement, a sensitive ϵ2 could lead to optimization instability or poor convergence during training. Can the authors provide a cross-task/model validation results to prove that 0.4 is a stable "universal" constant rather than a dataset/model-specific tuned value?


2. Definition of Visual-Related Tokens (V_{vis})

- Could the authors details how are the visual-related token set defined? Specifically, how was the vocabulary filtered to ensure that abstract or context-dependent visual terms are not omitted?


3. Evaluation Protocol (Zero-Shot vs. Fine-Tuning)

- Can the authors confirm whether the model is trained exclusively on the caption generation task and subsequently evaluated on discriminative tasks (e.g., POPE, MME) without any further fine-tuning?


4. Statistical Significance of Results

- In Table 2, while AFS reports the best performance across three benchmarks, the margins in some cases appear narrow. To justify the claim of superiority, the authors are encouraged to provide statistical significance tests (e.g., p-values or confidence intervals over multiple runs) to distinguish these gains from stochastic variance.


Other Comments (Writing and Technical Notations)
- Equation (4) Completeness: In Eq. (4), it appears that the input instruction/query (e.g., "describe the image") is missing from the conditional probability notation. For clarity and mathematical rigor, the full context should be included.

- Redundancy in Write-up: There is significant repetition of content between the Introduction and Section 3. The authors should streamline these sections, ensuring the Introduction focuses on high-level motivation while Section 3 focuses on the technical implementation details. This would improve the overall readability and "flow" of the paper.

**Limitations:**

Yes

**Strengths And Weaknesses:**

Strengths
- Principled Internal Supervision: The core contribution—leveraging layer-wise discrepancy as an intrinsic reward signal—is a efficient and reasonable. By framing hallucination as a token-level distributional alignment problem and combining with visual-selective reward, the work addresses the root cause of "language-prior dominance" in the final layers of LVLMs.

- Comprehensive Reward Design: The three-pillar framework (Reward Gating, Token Classification, and Distributional Consistency) is technically sound. Specifically, the use of GRPO allows for stable fine-tuning without the memory overhead of a separate critic model, which is a significant practical advantage for large-scale LVLMs.

- The effectiveness of AFS across both discriminative and generative benchmarks suggests that the "anchor-final" discrepancy is a universal phenomenon in transformers. The reported improvements in cross-domain generalization indicate that the model is learning a fundamental visual-grounding principle rather than over-fitting to specific dataset biases.

Weaknesses
- Sensitivity to Threshold Hyperparameters: While the design is logical, the framework appears to rely on sensitive hyperparameters, such as ϵ2​ in the distributional consistency reward. The paper would be strengthened by a sensitivity analysis on different training dataset and architecture to demonstrate how performance fluctuates across different values of ϵ2​ and whether these thresholds need to be re-tuned for different model architectures (e.g., LLaVA vs. Qwen-VL).

- Heuristic Dependency in Token Identification: The "Visual-selective reward gating" currently relies on a pre-defined list of visual-related tokens. This is a somewhat ad hoc solution that might limit the model's ability to handle abstract or complex visual concepts not captured in the list. As the reviewer notes, leveraging an LLM to dynamically identify these tokens would be a more scalable approach, and the current reliance on a static list should be acknowledged as a limitation.

---

> ### Author Rebuttal · Authors · 2026-03-30
>
> **W1 and Q1: Robustness of Threshold $\epsilon_2$ and Training Stability**
>
> We appreciate this important concern regarding the sensitivity of $\epsilon_2$ and its potential impact on training stability. Following your suggestion, we evaluate $\epsilon_2$ across both a different training dataset (Flickr30k) and a different architecture (InternVL2.5-4B).
>
> | $\epsilon_2$ | Flickr30k |           |          | InternVL2.5-4B |           |          |
> | ------------ | --------- | --------- | -------- | -------------- | --------- | -------- |
> |              | CHAIR_S ↓ | CHAIR_I ↓ | Recall ↑ | CHAIR_S ↓      | CHAIR_I ↓ | Recall ↑ |
> | Vanilla      | 28.8      | 9.1       | 64.3     | 38.0           | 8.0       | 70.5     |
> | 0.3          | 25.4      | 7.8       | 64.2     | 33.6           | 7.8       | 69.4     |
> | 0.4          | **25.2**  | **7.7**   | **64.9** | 33.4           | 7.5       | 70.3     |
> | 0.5          | **25.2**  | 8.2       | 64.5     | **32.2**       | **7.4**   | 70.4     |
> | 0.6          | 25.8      | 8.3       | 64.8     | 32.8           | **7.4**   | **70.6** |
>
> We further analyze the distribution of maximum anchor layer confidence and observe that it is highly concentrated toward the extremes, with only a small fraction of tokens falling into the intermediate range where  $\epsilon_2$ takes effect:
>
> | Maximum Anchor Confidence | Proportion |
> | ------------------------- | ---------- |
> | [0.0, 0.3)                | 36.64%     |
> | [0.3, 0.6)                | 14.55%     |
> | [0.6, 1.0)                | 48.81%     |
>
> This explains the observed stability, as only a small subset of tokens is affected by changes in $\epsilon_2$ . Overall, $\epsilon_2$ is not a sensitive or dataset/model-specific hyperparameter, as performance remains stable across a broad range of values under both dataset and architecture changes.
>
> **W2 and Q2: Design and Coverage of $V_{vis}$ and All Concerns Related to $V_{vis}$**
>
> Thank you for raising this important question. We appreciate your concern on the coverage of $V_{vis}$, Reviewer Mt9R’s question on classification sensitivity and stability, and Reviewer 2Si6’s concern on ambiguity and subword effects. We apologize that the motivation behind the design of $V_{vis}$ was not clearly explained in the paper, and provide a unified clarification below.
>
> Our goal is **not to perfectly identify all visual descriptive tokens**, but to provide a **simple and low-cost gating mechanism** to identify tokens that should be assigned reward signals.
>
> In this context, the design is robust for three reasons:
>
> 1. **It serves as a lightweight gating mechanism rather than a primary driver of performance**:
>
>    $V_{vis}$ only determines where rewards are applied, while the primary learning signal comes from anchor–final signals. Therefore, our method is not sensitive to precise token classification.
>
> 2. **Ambiguity and omissions have limited impact**:
>    As noted by Reviewer 2Si6 and Reviewer Mt9R, some tokens may be ambiguous or context-dependent. In practice, such cases affect only a small fraction of tokens, and resulting noise has negligible influence on the overall optimization.
>
> 3. **Semantically meaningful tokens dominate learning signals**:
>    Even with subword tokenization, tokens that carry clear visual semantics dominate generation and reward assignment, while fragmented or noisy tokens contribute minimally.
>
> We strongly appreciate your suggestion of dynamically identifying visual descriptive tokens using LLMs, as well as the similar insight on contextualized generation from Reviewer 2Si6. Such approaches can naturally address ambiguity and context dependence, and we believe both are reasonable choices, reflecting trade-offs between stability, adaptivity, and computational efficiency.
>
> Overall, our design adopts a lightweight approach, and the resulting noise does not materially affect the optimization process, reflecting a practical trade-off in favor of efficiency and scalability.
>
> **Q3: Evaluation Protocol**
>
> Yes, the model is trained exclusively on the caption generation task, and discriminative benchmarks are evaluated without further fine-tuning. This design ensures that the observed improvements reflect improved faithfulness to visual evidence rather than task-specific adaptation.
>
> **Q4: Statistical Significance of Results**
>
> Following your suggestion, we report mean ± std over 3 runs under stochastic decoding (temperature = 0.7, top_p = 0.9).
>
> | Method     | POPE-F1          | AMBER-F1         |
> | ---------- | ---------------- | ---------------- |
> | Vanilla    | 84.36 ± 0.19     | 87.73 ± 0.09     |
> | DeCo       | 84.64 ± 0.17     | 87.87 ± 0.09     |
> | VCD        | 85.69 ± 0.07     | 87.23 ± 0.05     |
> | AFS (Ours) | **85.85** ± 0.06 | **88.17** ± 0.05 |
>
> These results show that the improvements are consistent across runs.
>
> We sincerely thank the reviewer for providing suggestions that improve the mathematical precision as well as the overall flow of the paper.

---

> > ### Author Rebuttal · Reviewer_1vN2 · 2026-04-01
> >
> > The authors has addressed my concerns, I don't have specific questions on my initial weakness raised. I am leaning toward accepting this paper, but i will carefully consider the concerns raised from fellow reviewers.

---

> > > ### Author Response · Authors · 2026-04-03
> > >
> > > We sincerely thank the reviewer for the constructive feedback and for acknowledging that our responses addressed your initial concerns. We greatly appreciate your careful consideration and the attention given, as also reflected in the positive responses from other reviewers. We look forward to your final decision and appreciate the time you spent reviewing our work.

---

### Official Review · Reviewer_Mt9R · 2026-03-13

**Soundness:** 3
**Presentation:** 3
**Significance:** 3
**Originality:** 3
**Overall Recommendation:** 4
**Confidence:** 3

**Summary:**

The paper addresses the critical issue of hallucinations in Large Vision-Language Models (LVLMs), where models prioritize linguistic priors over visual evidence. The authors propose **Anchor-Final Self-Supervision**, a training-time framework that leverages internal model signals—specifically the discrepancy between intermediate "anchor" layers and the final output layer—as a self-supervised reward. Results across generative (CHAIR, AMBER) and discriminative (POPE, MME) benchmarks show that AFS significantly reduces hallucinations while maintaining or even slightly improving content recall.

**Compliance With Llm Reviewing Policy:**

Affirmed.

**Key Questions For Authors:**

（1）How sensitive is the final performance to the accuracy of the "Visual vs. Non-Visual" token classifier? If the classifier misses key visual tokens, does the model default to its original hallucination-prone behavior?


（2） Since AFS encourages consistency with intermediate layers (which are more visually grounded), is there any risk of the model losing its ability to follow complex linguistic instructions that require high-level abstract reasoning found in the final layers?

**Limitations:**

Yes

**Strengths And Weaknesses:**

### **Strengths**

（1） The work is well-grounded in recent findings that hallucinations often arise in the later stages of decoding where language priors dominate. Bringing these "decoding-time" insights into a "training-time" optimization framework is a natural and effective progression.


（2） Unlike traditional methods that require costly external labels, human feedback, or ground-truth captions, AFS operates entirely on internal model dynamics. This makes it highly scalable.


（3） AFS achieves impressive reductions in hallucination metrics without the typical "recall trade-off" where models become overly concise to avoid errors.


### **Weaknesses**

（1） The method assumes access to a predefined set of visual-related token identifiers ($\mathcal{V}_{vis}$). While the appendix suggests using an LLM to classify these, the stability and quality of this classification across different languages or highly specific domains are not fully explored.

（2） Implementing layer-wise probability calculations and consistency rewards during training via GRPO adds overhead compared to standard Supervised Fine-Tuning (SFT). A more explicit discussion of the training wall-clock time and memory requirements would be beneficial.

---

> ### Author Rebuttal · Authors · 2026-03-30
>
> **Q2: Impact on Complex Instructions and Abstract Reasoning**
>
> We appreciate your thoughtful concern regarding the potential impact on instruction following and high-level reasoning.
>
> **Empirically**, AFS maintains or slightly improves performance on:
>
> * MME Cognition (Table 2), which evaluates multimodal reasoning and knowledge-intensive tasks.
> * MM-Vet (Table 10), which evaluates complex tasks, including spatial and mathematical reasoning.
>
> This indicates that high-level reasoning and instruction-following abilities are preserved.
>
> **Mechanistically**, this is due to two key design choices:
>
> * **Selective optimization**: rewards are applied only to visually descriptive tokens, leaving linguistic and reasoning-related tokens unconstrained.
> * **Guidance rather than restriction**: AFS encourages consistency with intermediate visual evidence, rather than constraining the final layer. This preserves the model’s ability to perform high-level reasoning while reducing hallucination.
>
> Additionally, improved consistency with visual evidence helps reduce incorrect premises during reasoning, which can further benefit high-level reasoning performance.
>
>
>
> **W2: Computational Cost and Efficiency**
>
> Thank you for raising this important and practical concern regarding computational overhead. The dominant cost in our framework comes from **GRPO**, which requires sampling multiple rollouts (K=12 by default). Compared to standard GRPO, AFS introduces only modest additional overhead:
>
> * **+7.8% wall-clock time**
> * **+3.4% peak memory**
>
> This overhead comes from exposing intermediate-layer representations during rollout and computing layer-wise probability for reward construction. Importantly:
>
> * Reward computation is **gradient-free**
>
> * No additional backpropagation or auxiliary networks are introduced beyond standard GRPO
>
> Therefore:
>
> * Compared to **decoding-time methods**, AFS incurs cost only during training and has **no inference overhead**
> * Compared to **GRPO-based methods**, the additional cost of AFS is minor relative to rollout sampling
>
> Overall, AFS achieves a favorable trade-off between efficiency and effectiveness, with most of the overhead coming from GRPO itself rather than our proposed components.
>
> | Method        | Time / Step (s) | Peak Memory (MB) |
> | ------------- | --------------- | ---------------- |
> | SFT           | 1.34            | 51355            |
> | Standard GRPO | 24.75           | 126993           |
> | AFS (Ours)    | 26.67           | 131349           |
>
>
>
> **Q1 and W1: Sensitivity and Stability of Visual-related Token Classifier**
>
> Thank you for this thoughtful question on the robustness of the visual-related token classifier. This concern is closely related to the design of $V_{vis}$ , and we kindly refer you to our response to Reviewer 1vN2 (W2 and Q2: Design and Coverage of $V_{vis}$ and All Concerns Related to $V_{vis}$) for details.

---

> > ### Author Rebuttal · Reviewer_Mt9R · 2026-04-01
> >
> > The authors have addressed my concerns. I keep my initial positive scores.

---

> > > ### Author Response · Authors · 2026-04-03
> > >
> > > We sincerely thank the reviewer for the positive scores and for acknowledging that our responses have addressed your concerns. We greatly appreciate your thoughtful feedback and careful consideration.

---

### Official Review · Reviewer_2Si6 · 2026-03-15

**Soundness:** 2
**Presentation:** 3
**Significance:** 2
**Originality:** 2
**Overall Recommendation:** 4
**Confidence:** 3

**Summary:**

This paper proposes AFS, a self-supervised training-time framework for reducing hallucinations in large vision-language models. The central hypothesis is that hallucinated visual tokens tend to receive weak support from intermediate “anchor” layers but are amplified by the final layer. Based on this hypothesis, the method applies reward gating only to visually descriptive tokens, classifies visual tokens as hallucinated or non-hallucinated using anchor-layer confidence and the anchor-final probability gap, and further introduces an anchor-final distributional consistency term to define rewards within GRPO fine-tuning. The model is trained on caption-free MSCOCO images and evaluated on both generative and discriminative hallucination benchmarks.

**Compliance With Llm Reviewing Policy:**

Affirmed.

**Final Justification:**

The authors tackled my concerns in weaknesses and questions. I encourage the authors to provide more discussion of limitations and practical evidence of the benefits on strong VLM baselines.

**Key Questions For Authors:**

- **(Q1)** Can the authors report what fraction of generated tokens are gated as visually descriptive, and how the final reward mass is distributed across gated versus non-gated tokens? This would clarify whether the reported gains come from a broad re-weighting of grounded content or from a relatively small subset of especially influential tokens.

- **(Q2)** How robust is the learned behavior to prompt variation at inference time? Since training appears to use only a single captioning prompt, it would be useful to know whether AFS still helps under shorter, longer, object-centric, and question-answering style prompts.

- **(Q3)** Can the authors provide a finer-grained error analysis of remaining failures, separated into object, attribute, relation, and counting errors, together with representative layer-wise probability traces?

**Limitations:**

The method depends on access to intermediate-layer probabilities during training, which may limit portability to closed-source LVLM APIs or to architectures whose training stacks do not expose the required internal states. Because the supervision signal is derived from the model’s own internals, systematic blind spots of the base model may be reinforced rather than corrected when the underlying visual evidence is consistently misread.

**Strengths And Weaknesses:**

### Strengths

- **(S1)** The paper studies an important and practically relevant problem, namely, visual hallucination in LVLMs, and it approaches this problem from a meaningful angle. The method is conceptually coherent, because its three components are tied to a single motivating hypothesis: hallucinated tokens are weakly supported by intermediate visual evidence yet amplified in later layers by language priors.

- **(S2)** The empirical section has reasonable breadth and contains some genuinely encouraging findings. In particular, the CHAIR improvements are non-trivial while recall is roughly preserved, and the paper also evaluates transfer to POPE, MME, AMBER, and MM-Vet, together with ablations over reward factors, anchor-layer choice, and reward sharpness.

### Weaknesses

- **(W1)** The central framing is stronger than what the implementation actually supports. The paper repeatedly presents AFS as a form of “token-level optimization,” yet the GRPO update ultimately relies on a sequence-level scalar reward obtained by aggregating token rewards. Therefore, the method is more accurately described as token-aware reward shaping for sequence-level RL fine-tuning.

- **(W2)** Several core mechanisms remain heuristic and are only partially validated. The hallucination classifier depends on fixed thresholds over anchor confidence and the sign of the anchor-final gap, while the predefined visual token set is produced offline by the same model family on tokenizer entries rather than contextualized generations. Since many tokenizer items are ambiguous or subword-level, this design may introduce self-confirming bias.

- **(W3)** The comparison protocol is not sufficiently strong for a training-time contribution. Although the paper positions itself against a broader training-time literature, the experiments compare only against decoding-time baselines. As a result, it remains unclear whether the gains should be attributed to the specific anchor-final self-supervision design or, more generally, to performing additional task-specific fine-tuning.

- **(W4)** The empirical support, presentation, and reproducibility are not yet at the level I would expect for acceptance. The CHAIR results are promising, but they are reported on only a limited validation subset. Moreover, the current draft still contains presentation artifacts and does not clearly quantify the extra compute and memory cost induced by intermediate-layer extraction and GRPO sampling, which makes the practical cost–benefit trade-off harder to assess.

---

> ### Author Rebuttal · Authors · 2026-03-30
>
> **Q1: Fraction of Gated Tokens and Reward Concentration**
>
> We report both tokenizer and training-time gating ratios over 1,024 images with 12,288 generations in GRPO. To reflect the English caption setting, we also report results on alphabetic tokens.
>
> | Token Type                    | Total Count | Visual Descriptive Count | Gated Ratio |
> | ----------------------------- | ----------- | ------------------------ | ----------- |
> | Tokenizer (all tokens)        | 151665      | 8890                     | 5.86%       |
> | Tokenizer (alphabetic tokens) | 70096       | 2416                     | 3.45%       |
> | Training (generated tokens)   | 1996018     | 134365                   | 6.73%       |
>
> This confirms that the improvements are driven by a small subset of gated tokens, rather than broad reweighting of the entire sequence.
>
>
>
> **Q2: Robustness to Prompt Variation.**
>
> We agree that prompt robustness is important. Although training uses a single captioning prompt, our evaluation covers multiple instruction prompts at inference (e.g., POPE, MME, AMBER, MM-Vet). We observe consistent improvements across all settings, which demonstrates the generalization ability of our approach and its robustness to prompt variation.
>
>
>
> **Q3: Finer-grained Error Analysis and Layer-wise Traces**
>
> Thank you for this valuable suggestion. We provide a finer-grained analysis of the 1,004 generative caption samples in AMBER, categorizing remaining failures into object, attribute, relation, and counting errors. We further present two representative examples (see corresponding images at: https://anonymous.4open.science/r/examples-B21E/) together with their layer-wise probability traces, where layer indices correspond to [3,6,9,12,15,18,21,24,27,28].
>
> | Error Type | Count | Ratio |
> | ---------- | ----: | ----: |
> | Object     |    70 | 68.6% |
> | Attribute  |    20 | 19.6% |
> | Relation   |    11 | 10.8% |
> | Counting   |     1 |  1.0% |
>
> | Image ID | Generated Description                                        | Hallucinated Token | Type      | Layer-wise Probability Trace                                 |
> | -------- | ------------------------------------------------------------ | ------------------ | --------- | ------------------------------------------------------------ |
> | 181      | The image shows two cows in a grassy field... The background consists of green grass and some trees... | trees              | Object    | [0.096, 0.083, 0.074, 0.042, 0.018, 0.015, 0.013, 0.007, 0.012, 0.266] |
> | 253      | The image shows a zebra grazing... with some scattered yellow flowers... | yellow             | Attribute | [0.118, 0.082, 0.044, 0.035, 0.029, 0.013, 0.004, 0.003, 0.002, 0.625] |
>
> These results show that failures are dominated by object-level hallucinations and tend to be amplified at later layers, consistent with our motivation and overall design goal.
>
>
>
> **W1: Method Framing**
>
> We sincerely thank the reviewer for the valuable comments on method framing, which help improve the precision of our description.
>
>
>
> **W2: Heuristic mechanisms and Validation**
>
> Thank you for raising this important concern. For mechanisms including hallucination classifier thresholds and visual token set, detailed analyses are provided in our responses to Reviewer 1vN2 (W1 and Q1: Robustness of Threshold $\epsilon_2$ and Training Stability; W2 and Q2: Design and Coverage of $V_{vis}$ and All Concerns Related to $V_{vis}$), showing that the proposed mechanisms are robust and provide reliable training signals.
>
> **W3: Comparison Protocol**
>
> We kindly refer you to our responses to Reviewer caJo (Q1 and W3: Comparison with Training-Time Baseline), showing that the gains are not due to generic training-time adaptation but are attributable to anchor–final signal optimization.
>
> **W4: Robustness Across Validation Subsets and Computational Cost**
>
> Our original evaluation follows common practice in prior work including Deco[1] for consistency.
>
> To address your concern about limited validation subset, we conduct experiments with 5 random seeds, each corresponding to a different subset of 500 images. We report the aggregated results (mean ± std) across these subsets:
>
> | Method  | CHAIR_S ↓        | CHAIR_I ↓       | Recall ↑         | Len           |
> | ------- | ---------------- | --------------- | ---------------- | ------------- |
> | Vanilla | 28.28 ± 0.47     | 8.90 ± 0.17     | 64.60 ± 1.04     | 108.44 ± 0.77 |
> | Ours    | **24.08** ± 0.88 | **7.56** ± 0.23 | **64.80** ± 1.10 | 106.36 ± 0.62 |
>
> These results show consistent improvements, indicating robustness across different validation subsets.
>
> [1]. MLLM can see? Dynamic Correction Decoding for Hallucination Mitigation. ICLR 2025.
>
> For compute and memory cost, please refer to our response to Reviewer Mt9R (W2: Computational Cost and Efficiency) for additional quantitative results, which validate that the additional overhead is modest (less than 8%) relative to standard GRPO.

---

> > ### Author Rebuttal · Reviewer_2Si6 · 2026-04-04
> >
> > Thanks for the detailed response. I believed the authors nearly tackled my concerns and decided to raise my score to 4. Meanwhile, more discussions of the mentioned limitations could be further provided. Also, the authors should provide more practical evidence or cases on strong VLM baselines (e.g., based on Qwen3-VL or close-source VLM models) to show be practical benefits of this work.

---

> > > ### Author Response · Authors · 2026-04-06
> > >
> > > Thank you for the constructive follow-up and for raising your score! We are encouraged to see that our responses addressed most of your concerns, and we sincerely appreciate your careful evaluation and insightful suggestions, which helped us further strengthen both the empirical evidence and discussion of limitations.
> > >
> > > Following your valuable suggestion, we conducted experiments on Qwen3-VL-8B:
> > >
> > > | Method  | CHAIR_S ↓        | CHAIR_I ↓       | Recall ↑     | Len           |
> > > | ------- | ---------------- | --------------- | ------------ | ------------- |
> > > | Vanilla | 43.80 ± 1.46     | 9.54 ± 0.24     | 71.78 ± 1.34 | 241.68 ± 0.89 |
> > > | Ours    | **37.40** ± 1.21 | **8.32** ± 0.54 | **71.90** ± 1.29 | 242.90 ± 0.58 |
> > >
> > > This results further support the practical effectiveness of our method on stronger VLM.
> > >
> > > We also agree that our method depends on access to intermediate-layer probabilities, and is therefore primarily applicable to open-source LVLMs or architectures that expose internal states, and we will explicitly clarify this applicability in the paper.
> > >
> > > Regarding the concern that supervision signal derived from model internals may reinforce systematic blind spots, empirically, AFS consistently improves hallucination metrics across different LVLMs, indicating its effectiveness beyond any specific model. More broadly, for base models exhibiting different hallucination patterns, our method provides a perspective for analyzing the layer-wise probability trajectories of hallucinated tokens and for designing tailored optimization objectives that adapt to each model’s characteristics. This perspective offers a principled approach to mitigating hallucinations across diverse LVLM architectures, and we hope this provides reassurance regarding the potential of our method in mitigating such blind spots.
> > >
> > > We sincerely thank you again for your valuable suggestions on method framing, strengthening empirical validation, and clarifying limitations. These comments have significantly improved the rigor and clarity of our work.

---

### Decision · Program_Chairs · 2026-04-30

**Decision:**

Accept (regular)

**Comment:**

The paper presents Anchor-Final Self-Supervision (AFS), a training-time framework that leverages internal layer-wise disagreement to mitigate hallucinations in large vision-language models. Reviewers agree that AFS is a principled use of model internals, with a GRPO fine-tuning schedule that preserves recall while substantially lowering CHAIR/AMBER hallucination scores. The authors convincingly demonstrate cross-domain transfer and modest compute overhead.

The raised issues include the heuristic token-classification and the initial lack of training-time baselines. Authors added sensitivity studies, a comparison to MOCHa/HA-DPO, and clarified that the visual-token list is a lightweight gating mechanism rather than a strict hallucination detector. The remaining concerns, especially the narrow single-prompt training regime, are mitigated by consistent gains across diverse inference prompts.

Overall, the paper merits acceptance. It is recommended that the authors incorporate the reviewers' suggested revisions in the final version.